# UNION: Unsupervised 3D Object Detection using Object Appearance-based Pseudo-Classes

**Ted Lentsch**    **Holger Caesar**    **Dariu M. Gavrila**
Department of Cognitive Robotics
Delft University of Technology

## Abstract

Unsupervised 3D object detection methods have emerged to leverage vast amounts of data without requiring manual labels for training. Recent approaches rely on dynamic objects for learning to detect mobile objects but penalize the detections of static instances during training. Multiple rounds of (self) training are used to add detected static instances to the set of training targets; this procedure to improve performance is computationally expensive. To address this, we propose the method UNION. We use spatial clustering and self-supervised scene flow to obtain a set of static and dynamic object proposals from LiDAR. Subsequently, object proposals' visual appearances are encoded to distinguish static objects in the foreground and background by selecting static instances that are visually similar to dynamic objects. As a result, static and dynamic mobile objects are obtained together, and existing detectors can be trained with a single training. In addition, we extend 3D object discovery to detection by using object appearance-based cluster labels as pseudo-class labels for training object classification. We conduct extensive experiments on the nuScenes dataset and increase the state-of-the-art performance for unsupervised 3D object discovery, i.e. UNION more than doubles the average precision to $38.4$. The code is available at `github.com/TedLentsch/UNION`.

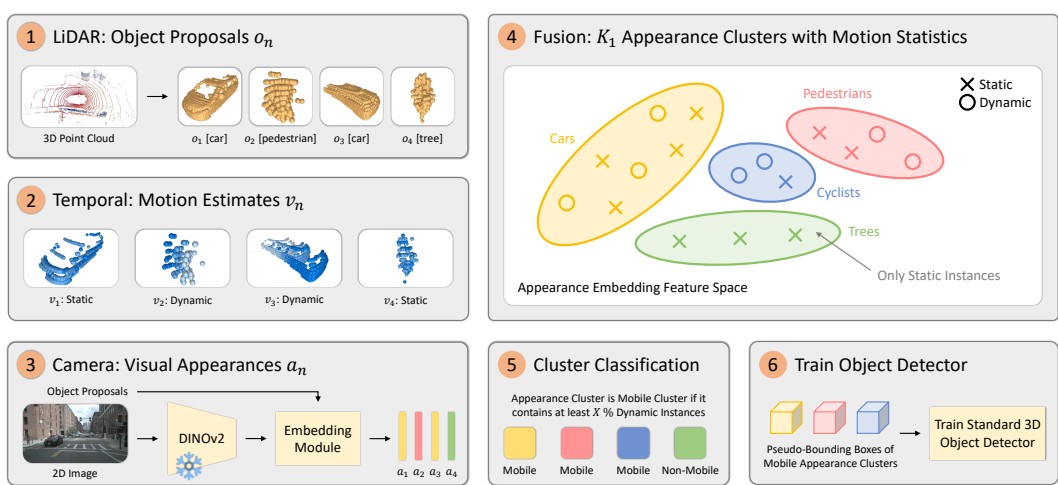

Figure 1: **UNION discovers mobile objects (e.g. cars, pedestrians, cyclists) in an unsupervised manner by exploiting LiDAR, camera, and temporal information** *jointly*. The key observation is that mobile objects can be distinguished from background objects (e.g. buildings, trees, poles) by grouping object proposals with similar visual appearance, i.e. clustering their appearance embeddings, and selecting appearance clusters that contain at least $X$ % *dynamic* instances.

38th Conference on Neural Information Processing Systems (NeurIPS 2024).

# 1 Introduction

Object detection is one of the core tasks of computer vision, and it is integrated into the pipeline of many applications such as autonomous driving [11], person re-identification [30], and robotic manipulation [35]. During the past decade, the computer vision community has made tremendous progress detecting objects, especially learning-based methods. These supervised methods rely on manual annotations, i.e. each object instance is indicated by a bounding box and a class label. However, a massive amount of labeled training data is usually required for training those models, while labeling is expensive and laborious. This raises the question of how object detection models can be trained without direct supervision from manual labels.

Unsupervised object detection is a relatively unexplored research field compared to its supervised counterpart. For camera images, recent work [4, 21] shows that the emergent behavior of models trained with self-supervised representation learning can be used for object discovery, i.e. object localization without determining a class label. The behavior implies that the learned features of those models contain information about the semantic segmentation of an image, and thus, they can be used to distinguish foreground from background. Consequently, the extracted coarse object masks are used to train 2D object detectors [22, 26]. Although these methods perform well for images depicting a few instances with a clear background, they fail to achieve high performance for images with many instances, such as autonomous driving scenes [27]. In these scenes, instances are close to each other and, as a result, are not directly separable using off-the-shelf features.

On the other hand, spatial clustering is the main force that drives 3D object discovery [27, 32]. In contrast to images, separating objects spatially is relatively easy in 3D space, but differentiating between clusters based on shape is challenging because of the density of the data (e.g. sparse LiDAR point clouds). Hence, temporal information is often exploited to identify dynamic points that most likely belong to mobile objects such as cars and pedestrians. In this context, we define mobile objects as objects that have the potential to move. Consequently, objects such as buildings and trees are considered non-mobile classes. The discovery of static foreground instances (e.g. parked cars and standing pedestrians) is usually achieved by performing self-training. Self-training is based on the assumption that a detector trained on dynamic objects has difficulty discriminating between the static and dynamic versions of the same object type. As a result, when such a detector is used for inference, it will also detect many static instances. The predicted objects are then used for retraining the detector, i.e. self-training, which is repeated multiple times until performance converges. Figure 2a gives a schematic overview of LiDAR-only methods for unsupervised 3D object discovery.

A significant drawback of iterative self-training is that training takes significantly longer due to the many rounds, e.g. sequentially training 5-10 times [2, 32], and there may be a confirmation bias, i.e. incorrect predictions early on can be reinforced. Moreover, we hypothesize that *training with only*

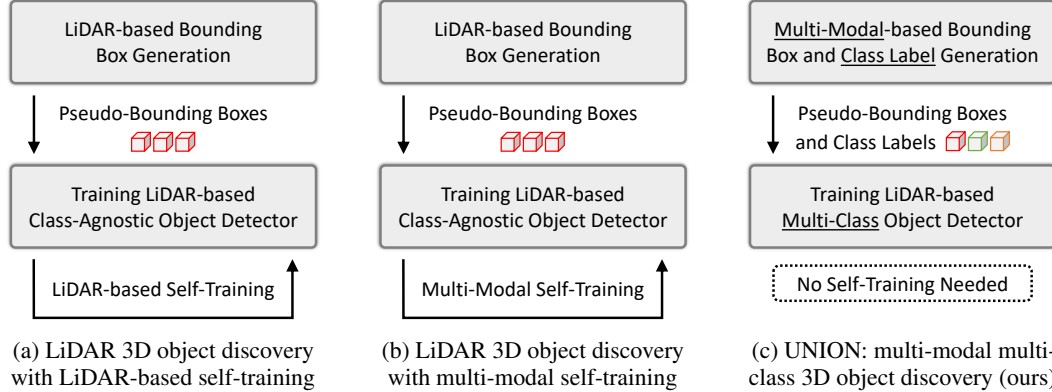

(a) LiDAR 3D object discovery with LiDAR-based self-training

(b) LiDAR 3D object discovery with multi-modal self-training

(c) UNION: multi-modal multi-class 3D object discovery (ours)

Figure 2: **Comparison of the various designs for unsupervised 3D object discovery. (a)** Most object discovery methods exploit LiDAR to generate pseudo-bounding boxes and use these to train a detector in a class-agnostic setting followed by self-training. **(b)** Wang et al. [27] generate pseudo-bounding boxes similar to (a) but alternate between training a LiDAR-based detector and a camera-based detector for self-training. **(c)** We use multi-modal data for generating pseudo-bounding boxes and pseudo-class labels, and train a multi-class detector without requiring self-training.

*dynamic objects degrades the final detection performance because (1) there is inconsistency during training and (2) the data distribution of static and dynamic objects differs.* The training inconsistency entails that the detection of static objects is penalized during training, i.e. they are considered false positives, while the objective of object discovery methods is to detect both static and dynamic objects. Besides, static and dynamic objects are sensed differently as they typically occur at different positions with respect to the sensors, and as a result, their data distribution differs (e.g. point distribution).

We argue that multi-modal data should be used *jointly* for unsupervised 3D object discovery as each modality has its own strengths, e.g. cameras capture rich semantic information and LiDAR provides accurate spatial information. Existing work [27] does use multi-modal data for unsupervised object discovery but not *jointly*. As shown in Figure 2b, the training procedure consists of two parts: (1) training with LiDAR-based pseudo-bounding belonging to dynamic instances and (2) multi-modal self-training to learn to detect static and dynamic objects. However, Wang et al. [27] ignore the fact that both modalities can be used at the same time for creating pseudo-bounding boxes.

Therefore, we propose our method, **UNION** (**un**supervised multi-modal 3D object detec**tion**), that exploits the strengths of camera and LiDAR *jointly*, i.e. as a *union*. We extract object proposals by spatially clustering the non-ground points from LiDAR and leverage camera to encode the visual appearance of each object proposal into an appearance embedding. Subsequently, we exploit the appearance similarity between static and dynamic foreground objects for discriminating between static foreground and background instances (see Figure 1). Finally, the identified objects and their appearance embeddings are used to generate pseudo-bounding boxes and pseudo-class labels, which can be used to train existing 3D object detectors in an unsupervised manner using their original training protocol. Figure 2c shows the high-level concept of our method with the novelties underlined.

Our contributions are twofold. **1.** We propose *UNION*, the first method that exploits camera, LiDAR, and temporal information *jointly* for training existing 3D object detectors in an unsupervised manner. We reduce training complexity and time by avoiding iterative training protocols. We evaluate our method under various settings and set a new state-of-the-art (SOTA) for object discovery on the nuScenes [3] dataset. **2.** Rather than training a detector to only distinguish between foreground and background, we extend 3D object discovery to *multi-class* 3D object detection. We utilize the appearance embeddings from UNION to create pseudo-class labels and train a multi-class detector.

## 2 Related work

Here, we review the work most related to unsupervised multi-modal 3D object detection.

**Unsupervised 2D object discovery** methods learn to detect objects in images without manual annotations and usually only use camera information. In general, heuristics are utilized to distinguish foreground from background. Some methods [10, 12, 13, 23] perform co-segmentation, which is the problem of *simultaneously* dividing multiple images into segments corresponding to different object classes. These methods rely on strong assumptions about the frequency of the common objects and the required computation scales quadratically with the dataset size.

Table 1: **Overview of existing methods for unsupervised 3D object discovery.** In the modality column, *L* and *C* are abbreviations for *LiDAR* and *camera*, respectively. Kickstart indicates what object types are used for the first round of training, i.e. the training before the self-training, and *S* and *D* are abbreviations for *static* and *dynamic* objects, respectively. [†]These methods rely on repeated traversals of the same location for extracting dynamic objects from the scene.

| Method | Year | Modality | Kickstart | Key Novelty | Code |
|---|---|---|---|---|---|
| MODEST [32] | 2022 | L | D[†] | LiDAR-based self-training | ✓ |
| Najibi et al. [20] | 2022 | L | D | Spatio-temporal clustering | ✗ |
| OYSTER [33] | 2023 | L | S+D | Near-long range generalization | ✗ |
| DRIFT [17] | 2023 | L | D[†] | Heuristic-based reward function | ✓ |
| CPD [29] | 2024 | L | S+D | Prototype-based box refinement | ✓ |
| LISO [2] | 2024 | L | D | Trajectory optimization | ✓ |
| LSMOL [27] | 2022 | L+C | D | Multi-modal self-training | ✗ |
| UNION (ours) | 2024 | L+C | S+D | Appearance-based pseudo-classes | ✓ |

In contrast, recent advances in representation learning [4, 21] show that the features of models trained with self-supervised learning contain implicit information about the semantic segmentation of a camera image. As a result, several works [16, 22, 25, 26, 28] exploit these models to get pseudo-bounding boxes and segmentation masks for training unsupervised detection models. LOST [22] leverages the features of the self-supervised method DINO [4] to extract foreground instances, and the clustered foreground instances are used to train a multi-class object detector. TokenCut [28] revisits LOST and proposes to build a graph of connected image regions to segment the image. CutLER [25] extends TokenCut for detecting multiple objects in a single image.

We also exploit a model trained with self-supervised learning [21]. However, we use the features to embed the appearance of 3D spatial clusters instead of differentiating between background and foreground in a 2D camera image. LOST extends unsupervised 2D object discovery to detection by creating pseudo-class labels using the CLS token of DINO for each object. Analogously, we extend the 3D case from discovery to detection, but our semantic embedding strategy differs from LOST. We extract camera-based features for a spatial cluster using its LiDAR points and aggregate these features to obtain the appearance embedding for a cluster.

**Unsupervised 3D object discovery** methods often only use LiDAR data to detect objects in 3D space. Table 1 provides an overview of the related work and is further explained below. MODEST [32] exploits repeated traversals of the same location for extracting dynamic objects from a LiDAR point cloud and uses self-training to also learn to detect static objects. DRIFT [17] also relies on repeated traversals for discovering dynamic objects but has a different self-training protocol. It uses a heuristic-based reward function to rank the predicted bounding boxes, and the highest-ranked bounding boxes are used to retrain the detector. The need for repeated traversals imposes a strong requirement on the data collection and limits the amount of data that can be used from existing datasets. Similarly to DRIFT, CPD [29] uses multi-class bounding box templates based on size information from Wikipedia to obtain a set of bounding boxes used for bounding box refinement, i.e. re-size and re-location.

Najibi et al. [20] and LISO [2] use self-supervised scene flow estimation in combination with tracking to detect dynamic objects, and train a detector using these objects. On the other hand, OYSTER [33] extracts object clusters close to the LiDAR sensor and uses the translation equivariance of a CNN in combination with data augmentation to learn near-range to long-range generalization for discovering objects. As a result, OYSTER can detect both static and dynamic objects. However, OYSTER cannot distinguish foreground objects from the background, and consequently, the performance suffers from the number of detected false positives. In contrast, LSMOL uses camera data to learn to detect static objects in its multi-modal self-training. It uses self-supervised scene flow estimation to identify dynamic objects. After that, it exploits the similarity of visual appearance between static and dynamic objects to learn to detect both during self-training.

Similarly to [2, 20, 27], we use self-supervised scene flow [15] for detecting dynamic objects instead of relying on repeated traversals of the same location such as MODEST and DRIFT. In contrast to existing methods, we use the camera-based appearance of dynamic objects to distinguish between background and static foreground objects before training the detector. As a result, we can directly discover static and dynamic objects without needing self-training. [17, 29] use multi-class bounding box templates to score and refine pseudo-bounding boxes. However, there is a fundamental difference between these methods and our method in how the pseudo-classes are used to supervise the detector. [17, 29] provide class-agnostic pseudo-bounding boxes for training detectors. In contrast, we use the appearance embeddings from UNION to create pseudo-class labels and train object classification. By doing this, we are the first to extend 3D object discovery to 3D object detection. Lastly, we do not assume any class-specific geometric prior (e.g. bounding box template) during training and inference.

**Self-supervised feature clusters** have been utilized for tasks related to unsupervised object detection. Drive&Segment [24] uses LiDAR segments in combination with self-supervised camera features to compute pixel-wise pseudo-labels for 2D unsupervised semantic segmentation (USS). The method constructs LiDAR-based 3D segments and computes an appearance embedding for each segment using DINO [4]. After that, pseudo-labels (i.e. cluster IDs) are obtained by clustering all appearance embeddings. CAUSE [14] also uses self-supervised features to obtain class prototypes for 2D USS.

Similarly to Drive&Segment, we compute an object appearance embedding for segments. However, we exploit appearance to distinguish foreground from background instances instead of assigning each camera pixel to one of the pseudo-classes. More specifically, we are the first to select and discard segments based on the fraction of dynamic instances belonging to their appearance clusters.

# 3 Method

This section introduces our method UNION for unsupervised 3D object detection of static and dynamic mobile class instances. Our work leverages recent advances in self-supervised learning to overcome the difficulty of distinguishing between static foreground and background instances in sparse 3D point clouds. First, we explain the task of unsupervised 3D object detection. After that, we describe the workings of UNION for learning to detect mobile objects *without* using manual labels.

## 3.1 Unsupervised 3D Object Detection

The task of 3D object detection is to detect, i.e. localize and classify, objects in 3D space. We consider upright 3D bounding boxes. Hence, it is assumed that roll and pitch are equal to zero. Each bounding box $b = (x, y, z, l, w, h, \theta)$ consists of the center position $(x, y, z)$, the length, width, and height $(l, w, h)$, and the heading $\theta$. In general, $C$ unique semantic classes $c_k$ of interest are defined for a detection setting, and each object instance is assigned to one of the classes.

In the case of supervised learning, all objects are labeled, and thus, they can be used as targets for training a learnable detector. However, for the unsupervised case, only raw data is available, which means that heuristics should be used to create pseudo-labels that replace manual labels. These pseudo-labels are an approximation of the real labels that would be obtained by carefully manually annotating the scenes. As a result, the performance of supervised training is the upper bound for the performance that can be achieved by using the *same* amount of data for training.

We aim to develop a framework for training existing 3D object detectors without relying on manual labels. Assume we have a mobile robot equipped with calibrated and time-synchronized sensors, including LiDAR, camera, GPS/GNSS, and IMU. Also, assume that LiDAR and the camera have an overlap in their field of view (FoV), i.e. parts of the environment are sensed by both sensors.

**Input.** The input of the framework consists of the multi-modal data collected during one or multiple traversals by the robot described above. Since this is raw data and no manual annotations are involved, such a dataset is easy to acquire. A traversal is a sequence of $T$ time steps, and for each time step $t$, a single LiDAR point cloud and $Q$ multi-view camera images are available. Let $P_t \in \mathcal{R}^{L \times 3}$ denote the $L$-point 3D LiDAR point cloud, and let $I_{q,t} \in \mathcal{R}^{H \times W \times 3}$ denote RGB image $q$ captured by camera $q$, $q \in Q$. $H$ and $W$ denote image height and width, respectively. The projection matrix is available for projecting 3D points on the 2D image plane for each camera. In addition, extrinsic transformations and ego-motion compensation can be used to transform data across sensor frames and time.

**Output.** The framework's output consists of a set of pseudo-bounding boxes and pseudo-class labels for each time step of all traversals. These pseudo-labels can be used to train existing 3D object detectors using their original protocol, where the only difference is that the targets used during training are pseudo-labels instead of manual labels. Let $\mathcal{B}_t$ and $\mathcal{C}_t$ denote the sets of pseudo-bounding boxes and pseudo-class labels for time step $t$, respectively. Both sets consist of $N$ pseudo-instances where each pseudo-instance is defined by a pseudo-bounding box $b_{n,t} = (x, y, z, l, w, h, \theta)$ and pseudo-class label $c_{n,t}$. Assume that there are $K$ pseudo-classes, i.e. $c_{n,t} \in \{1, ..., K\}$. There is only one pseudo-class for class-agnostic object detection, and thus $c_{n,t} = 1$ for all object instances.

## 3.2 Overview of UNION

The pipeline of UNION consists of two stages: (1) object proposal generation and (2) mobile object discovery. In Figure 1, these stages are represented by steps 1-3 and 4-5, respectively. The objective of the first stage is to generate a set of 3D object-like clusters and gather information about each cluster, i.e. the visual appearance and motion. This information is then used in the second stage to identify groups of mobile class instances and generate pseudo-labels. These labels are utilized to train existing object detectors (step 6 in Figure 1). The trained detector is used *identically* to the fully supervised case during inference. In Section 3.3, we explain the process of object proposal generation. After that, we describe the discovery of mobile objects in Section 3.4.

## 3.3 Object proposal generation

Here, we discuss the four components used for generating object proposals (steps 1-3 in Figure 1).

**Ground Removal.** The first step for generating object proposals is to extract the non-ground (ng) points $P_{ng,t}$ from all LiDAR point clouds $P_t$ as the non-ground points may belong to mobile objects. We assume that the ground is flat, and we fit a linear plane for each point cloud using RANSAC [9]. We use an inlier threshold of $5\,\mathrm{cm}$ for fitting the plane and consider all points that are more than $30\,\mathrm{cm}$ above the fitted ground plane as the non-ground points.

**Spatial Clustering.** The non-ground points are spatially clustered to get object proposals, i.e. 3D segments. To deal with the sparsity of the point clouds, we aggregate for each non-ground point cloud $P_{ng,t}$ the past and next $M$ non-ground point clouds. We set $M$ equal to $7$ to obtain an aggregated point cloud that is based on 15 scans. After aggregating the points clouds, HDBSCAN [19] is used to extract $N$ object proposals $o_n$. These object proposals can be part of the foreground as well as the background. We set the minimum cluster size and cluster selection epsilon to 16 points and $0.50\,\mathrm{m}$, respectively. Step 1 in Figure 1 illustrates the generation of these *class-agnostic* 3D object proposals.

**Motion estimation.** We estimate the motion status of the object proposals to determine whether each proposal is static or dynamic. The object proposals contain temporal information as the non-ground points from multiple time steps have been aggregated before the spatial clustering. In other words, the motion can be observed when the 3D points of an object proposal are split into different sets based on their time step, i.e. undoing the aggregation. This is shown by step 2 in Figure 1.

We estimate the motion of each proposal using a modified version of the SOTA self-supervised scene flow estimation method ICP-Flow [15]. We assume that mobile objects only move relative to the ground plane, so we limit scene flow estimation to 2D translation plus yaw rotation. A motion estimate $v_n$ is obtained for each object proposal by calculating the velocity magnitude of the estimated motion. We consider all object proposals with at least $0.50\,\mathrm{m/s}$ to be dynamic objects. As a result, we obtain the sets of static and dynamic object proposals $\mathcal{O}_t^S$ and $\mathcal{O}_t^D$ for time step $t$, respectively.

**Visual appearance encoding.** We observe that objects from the same semantic class look visually similar. Therefore, we aim to compute a camera-based encoding that can be used to search for visually similar-looking static objects using a reference dynamic object. We leverage the off-the-shelf vision foundation model DINOv2 [21] for encoding the camera images. This vision foundation model is part of a family of self-supervised learning algorithms that leverage the concept of knowledge distillation to train neural networks without requiring any labeled data.

We compute a feature map $F_{q,t} \in \mathbb{R}^{H_F \times W_F \times C_F}$ for each camera image $I_{q,t}$. Here, $H_F$, $W_F$, and $C_F$ indicate the feature map's height, width, and number of feature channels, respectively. Subsequently, we use our embedding module to compute a visual appearance embedding $a_n \in \mathbb{R}^{C_F}$ for each object proposal $o_n$. In the module, the LiDAR points of an object proposal $o_{n,t}$ are projected to the image plane, and we assign to each point $p \in \mathbb{R}^3$ a camera-based feature vector $f_p \in \mathbb{R}^{C_F}$ using the computed feature map $F_{q,t}$. After that, a single object proposal's feature vectors are averaged to obtain the visual appearance embedding $a_n$. This process is illustrated by step 3 in Figure 1.

### 3.4  Mobile object discovery

Current methods for 3D object discovery start the training of the detector with dynamic objects only, which means that static mobile objects such as parked cars serve as negatives. This causes an inconsistent supervision signal during neural network optimization as the shape of those static mobile objects can be very similar to dynamic mobile objects of the same class. As a result, the detector may learn to exploit the small differences in data distribution between static and dynamic mobile objects to be able to reduce the amount of false positives, i.e. static objects that are detected. The common strategy to improve detection performance is to perform expensive self-training in which static objects are added to the training targets. However, this also adds background instances, which lowers the detector's precision.

We aim to create pseudo-labels for both static and dynamic foreground instances because (1) this gives a more consistent supervision signal during training, (2) it enlarges the set of targets for training, i.e. more samples, and (3) this removes the need for computationally expensive self-training. We are confident that the set with dynamic proposals consists of mobile objects but the set of static object proposals contains both background objects (e.g. houses, poles, and bridges) and static foreground objects (e.g. parked cars and standing pedestrians). On the other hand, static and dynamic mobile objects from the same class have a similar visual appearance, which is different from background

objects. Therefore, we exploit the visual appearance embeddings to search for static mobile objects in the set of static object proposals.

We cluster the appearance embeddings using the K-Means algorithm [18] to group visually similar-looking object proposals (see step 4 in Figure 1). The number of clusters $K_1$ is set to 20. The clustering is done for all proposals together to ensure that there are enough dynamic objects to be able to differentiate between mobile and non-mobile clusters. As the obtained clusters differ in size, we calculate the fraction of dynamic instances $X$ for each cluster to classify the clusters. We consider appearance clusters with at least $5\%$ dynamic object proposals as mobile clusters, while the other clusters are non-mobile clusters. This is illustrated by step 5 in Figure 1. Classifying the appearance clusters means that both the set of static object proposals and the set of dynamic object proposals are split into mobile and non-mobile object instances. As a result, we can discover static mobile objects without requiring self-training, and we are robust against objects falsely labeled as dynamic.

We discard all non-mobile objects to obtain the set of mobile objects $O_{mobile}$ and compute a pseudo-bounding box $b_i = (x, y, z, l, w, h, \theta)$ for each mobile object $o_i$ using the 3D bounding box fitting algorithm of MODEST [32]. These pseudo-bounding boxes can be used to train existing detectors (see step 6 in Figure 1). In contrast to existing methods that only do *class-agnostic* detection, we extend 3D object discovery to 3D object detection by clustering our mobile objects into $K_2$ appearance-based pseudo-classes using K-Means. As a result, we obtain a pseudo-class label $c_i \in \{1, ..., K_2\}$ for each object that can be used for training a *multi-class* detector, i.e. predicting bounding boxes and classes.

**Optional for experiment 2 (multi-class detection).** During inference, the appearances are exploited to match the pseudo-classes to $R$ real classes. First, appearance prototypes are determined for the classes. The $K_2$ appearance cluster centers from K-Means are used as appearance prototypes $a_k \in \mathbb{R}^{C_F}, k \in \{1, ..., K_2\}$ for the pseudo-classes, and for each real class, we use a single example image to compute real class appearance prototypes $a_r \in \mathbb{R}^{C_F}$. Subsequently, the cosine similarity is calculated between the prototypes of pseudo-classes and real classes, and each pseudo-class is assigned to the real class with the highest similarity with its appearance prototype. Note that this matching step requires negligible supervision as we do one-shot association during inference.

## 4 Experiments

This section describes the experimental setup, the used baselines, and the conducted experiments.

### 4.1 Experimental setup

**Dataset.** We evaluate our method on the challenging nuScenes [3] dataset. This is a large-scale autonomous driving dataset for 3D perception captured in diverse weather and lighting conditions across Boston and Singapore. It consists of 700, 150, and 150 scenes for training, validation, and testing, respectively. A scene is a sequence of $20\,\text{s}$, and is annotated with $2\,\text{Hz}$. Each frame contains one LiDAR point cloud and six multi-view camera images.

**Mobile object classes.** The nuScenes dataset has 10 detection classes. Eight of these relate to mobile objects, and we only use these for evaluation. Note that the labels of these mobile object classes

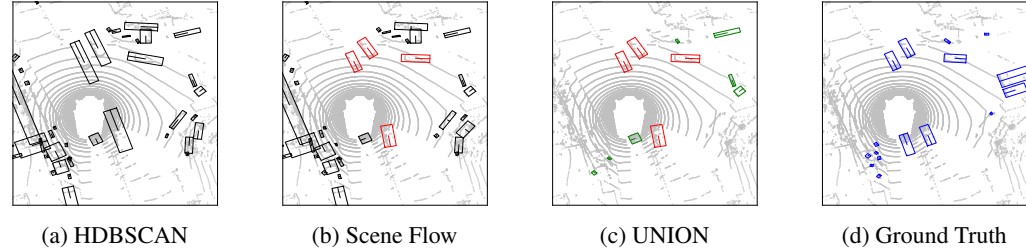

|          (a) HDBSCAN          |          (b) Scene Flow          |          (c) UNION          |          (d) Ground Truth          |

Figure 3: **Qualitative results for the UNION pipeline compared to the ground truth annotations.** (a) HDBSCAN (step 1 in Figure 1): object proposals (spatial clusters) in black. (b) Scene flow (step 2 in Figure 1): static and dynamic object proposals in black and red, respectively. (c) UNION: static and dynamic mobile objects in green and red, respectively. (d) Ground truth: mobile objects in blue.

Table 2: **Class-agnostic object detection on the nuScenes validation set.** Results are obtained by training CenterPoint [31] with the generated pseudo-bounding boxes. *L* and *C* are abbreviations for *LiDAR* and *camera*, respectively. Best performance in **bold**, and second-best is underlined. ST stands for *self-training*, which increases the computational cost of training. [†]Results taken from [2].

| Method | Labels | ST | AP ↑ | NDS ↑ | ATE ↓ | ASE ↓ | AOE ↓ | AVE ↓ |
|---|---|---|---|---|---|---|---|---|
| Supervised 1 % | Human | ✗ | 27.8 | 26.3 | 0.456 | 0.309 | 1.302 | 1.307 |
| Supervised 10 % | Human | ✗ | 61.2 | 56.7 | 0.255 | 0.221 | 0.462 | 0.455 |
| Supervised 100 % | Human | ✗ | 76.5 | 68.7 | 0.209 | 0.198 | 0.241 | 0.305 |
| HDBSCAN [16] | L | ✗ | 13.8 | 15.9 | **0.574** | 0.522 | 1.601 | 1.531 |
| OYSTER [29][†] | L | ✓ | 9.1 | 11.5 | 0.784 | 0.521 | 1.514 | - |
| LISO [1][†] | L | ✓ | 10.9 | 13.9 | 0.750 | **0.409** | 1.062 | - |
| UNION (ours) | L+C | ✗ | **38.4** | **31.2** | 0.589 | 0.497 | **0.874** | **0.836** |

are *not* used during training as our method UNION is fully unsupervised. For class-agnostic object detection (Section 4.3), the eight classes are grouped into a single object class for evaluation. On the other hand, for multi-class object detection (Section 4.4), we create three different classes, namely, (1) vehicle, (2) pedestrian, and (3) cyclist. The vehicle class combines the bus, car, construction vehicle, trailer, and truck classes, and the cyclist class combines the bicycle and motorcycle classes.

**Detector.** We use CenterPoint [31] for all our experiments. CenterPoint is a SOTA LiDAR-based 3D object detector. In contrast to traditional detection methods that generate bounding boxes around objects, CenterPoint uses a keypoint-based approach, detecting and tracking the centers of objects.

**Metrics.** The two main metrics that we consider are average precision (AP) [8] and the nuScenes detection score (NDS) [3], which are computed using the standard nuScenes evaluation protocol. AP is obtained by integrating the recall versus precision curve for recalls and precisions larger than 0.1, and averaging over match thresholds of $0.5 \, \text{m}$, $1.0 \, \text{m}$, $2.0 \, \text{m}$, and $4.0 \, \text{m}$. NDS is a weighted average of AP and five true positive errors, i.e. translation (ATE), scale (ASE), orientation (AOE), velocity (AVE), and attribute (AAE). We do all our experiments without attribute estimation similar to [2], and thus, we set the true positive errors for attribute to 1.0 by default.

**Baselines.** We compare against three unsupervised baselines for class-agnostic object detection, namely (1) HDBSCAN [19], (2) OYSTER [33], and (3) LISO [2]. In addition, we also compare to training CenterPoint using supervised learning with different subsets of the labels, i.e. 1 %, 10 %, and 100 %. We cannot compare to MODEST [32], Najibi et al. [20], DRIFT [17], CPD [29], and LSMOL [27]. MODEST and DRIFT need multiple traversals of the same location, which does not hold for all sequences in the nuScenes dataset. Furthermore, Najibi et al., CPD, and LSMOL did not release their code (at the time of submission) and did not provide performance on nuScenes.

Existing 3D object discovery methods cannot do *multi-class* object detection. Therefore, we use HDBSCAN in combination with class-based information as a baseline. Analogous to the class-agnostic setting, we also train CenterPoint using supervised learning for multi-class object detection.

### 4.2 Implementation

We use the framework MMDetection3D [5] for all our experiments and use their implementation of CenterPoint. More specifically, we use CenterPoint with pillars of $0.2 \, \text{m}$ as voxel encoder, do not use test time augmentation, and train for 20 epochs with a batch size of 4. All class-agnostic experiments are done without class-balanced grouping and sampling (CBGS) [34], while we do use CBGS for multi-class experiments to improve the performance of tail classes. The camera images were encoded with a large vision transformer (ViT-L/14) [7] trained using DINOv2 [21] with registers [6]. We used 8 NVIDIA V100 32 GiB GPUs for conducting the experiments. The hyperparameters of UNION were tuned visually on ten frames of the nuScenes training split. We made our code publicly available.

### 4.3 Class-agnostic object detection

We evaluate the performance of UNION for class-agnostic object detection on the nuScenes validation split. As shown in Table 2, UNION outperforms all unsupervised baselines in terms of AP and NDS. The best-performing unsupervised baseline is HDBSCAN, and UNION achieves an AP of more than

twice the AP of HDBSCAN. Both OYSTER and LISO score significantly lower than UNION despite using tracking in combination with self-training to improve detection performance.

When UNION is compared to supervised training, it can be seen that we outperform training with $1\%$ of the labels, but we are still behind the performance of using $10\%$ of the labels. This indicates that the dataset size and the quality of the targets used for training significantly impact detection performance. This is especially true for fine-grained labels such as the orientation, as the orientation convention depends on the object type. For example, determining the orientation of a car is relatively easy as the length of the car is typically much larger than the width, while for pedestrians, both dimensions are roughly the same. As a result, unsupervised approaches have difficulty determining the correct orientation for objects such as pedestrians.

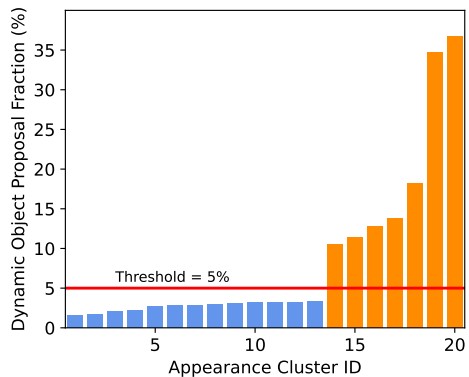

Figure 4: **The dynamic object proposal fractions of the visual appearance clusters.** We use a threshold of $5\%$ for selecting clusters.

Figure 3 provides qualitative results of the generated pseudo-bounding boxes for an example scene that was used for training the detector. The ground truth boxes are also shown. The figure shows the second sample from scene-1100 (training dataset). It can be seen that the scene flow can identify multiple dynamic objects, and the appearance clustering can discover static mobile objects, including vehicles and pedestrians, using those dynamic instances.

Figure 4 shows the percentage of dynamic object instances in the various appearance clusters, sorted by increasing percentage. We see a clear uptick at around $5\%$, which we use in the experiments as a threshold for distinguishing clusters of non-mobile objects (blue) versus mobile objects (orange).

**Ablation study I.** We now investigate the contribution of the various UNION components. Two intermediate representations of UNION can be used for generating pseudo-labels, namely (1) the output of the spatial clustering (step 1 in Figure 1) and (2) the output of motion estimation (step 2 in Figure 1). The output of the spatial clustering is identical to the HDBSCAN baseline in Table 2. The motion estimation differs from the spatial clustering output in that the points of dynamic object proposals are corrected for the estimated motion. The results are shown in Table 3 for different component-based pseudo-labels. As can be seen, appearance clustering is the main component that improves performance. This aligns with our expectation that appearance clustering can select mobile objects from the sets of static and dynamic proposals while discarding the background objects.

Table 3: **Class-agnostic object detection on the nuScenes validation set for different configurations of UNION.** Results are obtained by training CenterPoint [31] with the pseudo-bounding boxes. Best performance in **bold**.

| Method | AP ↑ | NDS ↑ |
|---|---|---|
| HDBSCAN | 13.8 | 15.9 |
| + Motion Estimation | 14.1 | 20.0 |
| + Appearance Clustering | **38.4** | **31.2** |

**Ablation study II.** Table 4 compares UNION's class-agnostic 3D object detection performance for different camera encoders, namely DINOv2 [21] and I-JEPA [1]. DINOv2 can process high-resolution images, such as the camera images from nuScenes. In contrast, I-JEPA can only process square-shaped images of a maximum of 448 by 448 pixels. As a result, the obtained feature maps from I-JEPA have a lower resolution than the ones from DINOv2. The table shows that UNION with DINOv2 outperforms UNION with I-JEPA by 15.6 in AP and 8.4 in NDS. Please note that our main paper contributions do not depend on specific canonical steps (e.g. image encoding, scene flow estimation). If better approaches become available, UNION can incorporate them.

Table 4: **Image encoder ablation study for UNION.** Best performance in **bold**.

| Method | AP ↑ | NDS ↑ | ATE ↓ | ASE ↓ | AOE ↓ | AVE ↓ |
|---|---|---|---|---|---|---|
| DINOv2 ViT-L/14 w/ registers [21] | **38.4** | **31.2** | 0.589 | 0.497 | **0.874** | **0.836** |
| I-JEPA ViT-H/16 [1] | 22.8 | 22.8 | **0.561** | **0.486** | 0.953 | 0.865 |

## 4.4 Multi-class object detection

As a second type of experiment, we perform multi-class object detection on nuScenes with 3 different semantic classes, namely vehicle, pedestrian, and cyclist, see Section 4.1. The mobile objects discovered by UNION are clustered into $K_2$ pseudo-classes. For evaluation, we have used three example instances from the nuScenes training dataset to associate each pseudo-class with one of the 3 real classes. This association procedure is described in Section 3.4. We also assign real classes to the class-agnostic predictions of HDBSCAN and UNION from Section 4.3. We do this by computing a prototype bounding box for each class, i.e. we select the bounding box with the median 2D area. Subsequently, we assign each class-agnostic bounding box to the real class of which the prototype bounding box has the highest intersection over union (IoU). We indicate this associating by *size prior*.

As shown in Table 5, UNION trained with 5 pseudo-classes performs the best and outperforms both HDBSCAN and UNION with the size prior in terms of AP and NDS. Differences in pedestrian detection performance mainly cause this. The vehicle detection performance of UNION-05pc is slightly worse. We observe that the cyclist performance is equal to zero for all configurations. From the pseudo-classes of multi-class UNION, there were 1, 1, 3, and 4 pseudo-classes assigned to the cyclist class for 5, 10, 15, and 20 pseudo-classes, respectively. Thus, it is not the case that all pseudo-classes are assigned to either the vehicle or pedestrian class. However, the nuScenes evaluation protocol only integrates the precision-recall curve for precision and recall larger than $0.1$. Therefore, we also evaluated without clipping the precision-recall curve as shown in the last column of Table 5. The results show that UNION-20pc significantly outperforms the baselines.

Table 5: **Multi-class object detection on the nuScenes validation set.** Results are obtained by training CenterPoint [31] with the generated pseudo-bounding boxes. *SP* stands for *size prior* and indicates that class-agnostic predictions from Table 2 are assigned to real classes based on their size. *UNION-Xpc* stands for UNION trained with $X$ pseudo-classes. *L* and *C* are abbreviations for *LiDAR* and *camera*, respectively. Best performance in **bold**, and second-best is underlined. [†]Without clipping the precision-recall curve, clipping is the default for nuScenes evaluation [3].

| | | Mobile Objects | | Vehicle | Ped. | Cyclist | Cyclist |
|---|---|---|---|---|---|---|---|
| Method | Labels | mAP ↑ | NDS ↑ | AP ↑ | AP ↑ | AP ↑ | AP[†] ↑ |
| Supervised 1 % | Human | 24.3 | 28.3 | 39.3 | 31.8 | 1.8 | 4.7 |
| Supervised 10 % | Human | 45.9 | 47.9 | 65.3 | 57.6 | 14.9 | 22.3 |
| Supervised 100 % | Human | 67.4 | 62.6 | 80.7 | 77.7 | 43.7 | 52.5 |
| HDBSCAN [16] + SP | L | 5.0 | 13.0 | 14.6 | 0.4 | **0.0** | 1.3 |
| UNION (ours) + SP | L+C | 12.7 | 19.7 | **34.8** | 3.4 | **0.0** | 1.6 |
| UNION-05pc (ours) | L+C | **24.0** | **24.0** | 30.3 | **41.6** | **0.0** | 0.8 |
| UNION-10pc (ours) | L+C | 19.9 | 21.7 | 27.3 | 32.5 | **0.0** | 0.5 |
| UNION-15pc (ours) | L+C | 18.5 | 21.2 | 25.7 | 29.9 | **0.0** | 0.4 |
| UNION-20pc (ours) | L+C | 17.9 | 21.7 | 23.7 | 29.9 | **0.0** | **4.2** |

## 5 Conclusion

We proposed UNION, the first framework that exploits LiDAR, camera, and temporal information *jointly* for generating pseudo-bounding boxes to train existing object detectors in an unsupervised manner. Rather than training an object detector to distinguish foreground and background objects, we perform *multi-class* object detection by clustering the visual appearance of objects and using them as pseudo-class labels. We reduce computational time by avoiding iterative training protocols and self-training. We evaluated our method under various settings and increase the SOTA performance for unsupervised 3D object discovery, i.e. UNION more than doubles the average precision to $38.4$.

**Limitations and future work.** A possible limitation of our work is that we make implicit assumptions about the occurrence frequency of objects by clustering the object proposals in the appearance embedding feature space. Mobile objects that are rare will likely be grouped with other objects, and as a result, these objects may be discarded when grouped with static background objects. Future work entails extending our method to better deal with these rare classes. In addition, the motion estimation could be based on radar detections as radars offer instant radial velocity estimation.

## Acknowledgments and Disclosure of Funding

This research has been conducted as part of the EVENTS project, which is funded by the European Union, under grant agreement No 101069614. Views and opinions expressed are, however, those of the author(s) only and do not necessarily reflect those of the European Union or European Commission. Neither the European Union nor the granting authority can be held responsible for them.

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
