# OpenReview forum: "UNION: Unsupervised 3D Object Detection using Object Appearance-based Pseudo-Classes"
_NeurIPS.cc/2024/Conference — NeurIPS 2024 poster_

### Official Review · Reviewer_JvDj · 2024-06-25

**Soundness:** 3
**Presentation:** 2
**Contribution:** 3
**Rating:** 6
**Confidence:** 4

**Summary:**

This work presents an unsupervised 3D object detection method named UNION, which exploits LiDAR, camera, and temporal information jointly for generating pseudo bounding boxes to train existing object detectors. In addition, the authors introduce an appearance-based clustering method to generate pseudo class labels and train the object detector in a multi-class fashion.

**Strengths:**

++ The method has significant improvement on the accuracy compared to existing unsupervised methods.

++ This method does not require the time-consuming multi-round self-training procedure.

**Weaknesses:**

-- There are not any qualitative results in the paper. Visualizations on the pseudo box/class generation and the comparative results would help readers understand the method more comprehensively.

-- Detailed analysis of failure cases would make this paper stronger. For example, in which scenarios UNION would generate bad pseudo labels, and how each step of UNION contributes to different failure cases.

-- Details about the HDBSCAN hyper-parameters are missing.

**Questions:**

-- Could the authors explain why OYSTER and LISO are not included in the multi-class experiments?

-- Will self-training help improve the detector trained with UNION pseudo labels?

-- For visual appearance encoding, DINOv2 usually can only get a lower-resolution feature map of the input image. How did the authors obtain point-wise features?

-- Which variant of DINOv2 is used for the experiments in this paper?

-- Could the authors provide the runtime analysis for each step of UNION?

**Limitations:**

Limitations are discussed in the paper.

---

> ### Author Rebuttal · Authors · 2024-08-07
>
> We thank you for your valuable feedback.
>
> **Qualitative results of pseudo-bounding box generation**
>
> We provide qualitative results of the intermediate outputs of the UNION pipeline and the final generated pseudo-bounding boxes **in Figure 2 in the rebuttal PDF**.
> This figure shows sample 2 from scene-1100, which is part of the nuScenes training dataset.
> It can be seen that the scene flow can identify multiple dynamic objects, and the appearance clustering can discover static mobile objects, including vehicles and pedestrians, using those dynamic instances.
>
> **UNION failure cases**
>
> The UNION pipeline consists of multiple components that each have their limitations, and as a result, UNION can fail to generate good pseudo-labels for some cases.
> The spatial clustering using HDBSCAN fails, for example, to generate correct clusters if multiple objects are close to each other, when objects are partially occluded, or when objects are far from the LiDAR (few points).
> This may be (partially) solved by using temporal tracking in combination with bounding box refinement.
> In addition, the self-supervised scene flow may fail to estimate a correct velocity when the spatial cluster contains few points.
> This also may be solved by using temporal tracking or other sensors such as radar.
> Finally, the parallax effect sometimes causes problems when computing the appearance embedding, e.g., LiDAR points are projected on the wrong object in the camera image.
> As a result, objects may be part of the wrong appearance cluster and, thus, may be falsely labeled as mobile or non-mobile.
> One failure case is shown at the bottom left of the UNION results in Figure 2.
> According to the ground truth, many pedestrians are located there, but UNION only discovers 2 of them (the bus stop partially occludes some and they have few LiDAR points).
>
> **HDBSCAN hyperparameters**
>
> We will add an appendix to the paper to list all the hyperparameters we used for UNION.
> For HDBSCAN, we used the implementation from scikit-learn, a Python module for machine learning built on top of SciPy.
> The minimum cluster size is set to 16 points, and the cluster selection epsilon is set equal to 0.50 meters, i.e., clusters are merged if they are within half a meter of each other.
>
> **OYSTER and LISO are not included in the multi-class experiment**
>
> The methods OYSTER [1] and LISO [2] are for unsupervised class-agnostic 3D object detection, i.e. they generate class-agnostic pseudo-bounding boxes that can be used for training existing detectors.
> So, their pseudo-bounding boxes cannot be used for training multi-class detectors as no (pseudo-)class labels are available.
> The same holds for HDBSCAN; therefore, we assigned class labels to the class-agnostic bounding boxes based on the size of the bounding boxes **(see Table 4 in the submission and Table 2 in the rebuttal PDF)**.
> However, the source code of OYSTER and LISO has not been released (yet), so we cannot use the size prior for these two methods.
> Consequently, we cannot compare multi-class UNION to 'OYSTER+Size prior' and 'LISO+Size prior' for the multi-class 3D object detection experiment.
>
> **Self-training with UNION**
>
> We have experimented with self-training for CenterPoint trained with the class-agnostic pseudo-labels from UNION.
> We used the following strategy for each self-training round: (1) predict with the trained CenterPoint on the training dataset from nuScenes, (2) filter the predictions using a score threshold, i.e. only keep the prediction if the predicted score is at least equal to the score threshold, and (3) train CenterPoint from scratch using the filtered predictions.
> We did this for three rounds for three different score thresholds, namely 0.10, 0.20, and 0.30.
> The average precision (AP) did not improve for any score thresholds.
> In addition, we noticed that the AP decreased after each round of self-training.
> Especially for score threshold 0.10, the performance drop was significant, i.e. the AP dropped to less than 20, while UNION achieved 38.4 **(see Table 1 in the rebuttal PDF)**.
> So, we did not experience any benefit from self-training after training with the UNION pseudo-labels.
> This is in line with the objective of UNION, discovering the static and dynamic mobile instances before training such that the computationally expensive self-training is not needed and that consistency during training enhances the detection performance (e.g. the detector is not penalized for detecting a parked car during training).
>
> **Point-wise camera-based features**
>
> The used large vision transformer has a stride of 14, i.e. the spatial resolution of the obtained feature map is in each dimension 14 times smaller.
> Each LiDAR point is first projected to the image plane.
> After that, the pixel coordinate is transformed to feature map coordinates (i.e. dividing by the stride), and then bilinear interpolation is used to get a camera-based feature vector for each LiDAR point so that the lower spatial resolution does not constrain us.
>
> **DINOv2 version**
>
> See global rebuttal text: 'Different image encoders'.
>
> **Runtime analysis of UNION**
>
> We have computed the runtime statistics for creating the class-agnostic pseudo-labels with UNION, which are used for training the detector (CenterPoint).
> We tested on a system with 2 Intel Xeon E5-2690 v4 CPUs (56 logical CPUs) and 8 NVIDIA V100 32GB GPUs.
> The training dataset of nuScenes consists of 700 sequences.
> In practice, we process the sequences in parallel in 8 threads.
> As a result, it takes 9, 19, 7, 9, and 2 hours for the ground removal, spatial clustering, motion estimation (scene flow), image encoding (DINOv2), and appearance embedding, respectively. After that, the appearance clustering is done for all sequences together and this takes 1 hour.
>
> [1] Zhang et al. (2023). Towards unsupervised object detection from lidar point clouds. In CVPR.
>
> [2] Baur et al. (2024). LISO: Lidar-only Self-Supervised 3D Object Detection. In ECCV.

---

> > ### Comment · Reviewer_JvDj · 2024-08-08
> >
> > Thank you for the responses. I have no further concerns.

---

### Official Review · Reviewer_Gw7r · 2024-07-09

**Soundness:** 3
**Presentation:** 3
**Contribution:** 2
**Rating:** 5
**Confidence:** 5

**Summary:**

This paper explores the challenge of unsupervised 3D object detection, introducing UNION. UNION leverages camera, LiDAR, and temporal information jointly to train 3D object detectors without relying on self-training. The approach demonstrates strong performance particularly on the nuScenes dataset. Additionally, the authors tackle unsupervised multi-class 3D object detection by clustering object appearance embeddings and employing these clusters as pseudo-class labels.

**Strengths:**

1. This paper studies a significant problem in unsupervised 3D object discovery.

2. The paper is clearly written, and its motivation is straightforward.

3. This method introduces visual embedding to enable certain static objects to be generated during object proposal generation.

4. This method has achieved good performance on nuScenes dataset.

**Weaknesses:**

1. Compared to prior work, the method's novelty is weak. The core innovation lies in leveraging self-supervised visual features to aid in object proposal extraction, thereby eliminating the need for self-training to acquire static vehicle data.
2. The core exploration of this work revolves around object appearance embedding. The current study only investigates the features of DINOv2. It remains to be explored how other self-supervised features perform in comparison, or whether combining different self-supervised methods could yield better results.
3. As for training the multi-class detector, I believe the choice of K is crucial. The authors use K=10 as default. It should conduct an ablation study on the impact of K.
4. As depicted in Table 2, the discussion centers around the challenges in orientation estimation. To enhance clarity, it would be beneficial to include detailed metrics such as ATE, AOE, etc.
5. In spatial clustering, when utilizing point cloud aggregation, how to deal with the dynamic objects, such as multiple moving cars whose point clouds may overlap.
6. As shown in Table 4, the AP for cyclists is always zero. Does this imply limitations of the method for detecting long-tail objects? What are some potential directions for improving detection of these less frequent objects in the future?

**Questions:**

See weakness.

**Limitations:**

The authors have discussed the limitations of the work.

---

> ### Author Rebuttal · Authors · 2024-08-07
>
> We thank you for your valuable feedback.
>
> **The method's novelty**
>
> UNION extracts object proposals by spatially clustering the non-ground points from LiDAR.
> Subsequently, the velocity of each object proposal is estimated using self-supervised scene flow, and the cameras are leveraged to encode its visual appearance into an object appearance embedding.
> These different types of information are then fused in a novel way to discover mobile objects in 3D space, such as vehicles, pedestrians, and cyclists, namely the visual appearances are clustered into $K_{1}$ appearance clusters and the fraction of dynamic object proposals per appearance cluster is calculated.
> The appearance similarity between static and dynamic mobile objects is utilized for discriminating between static mobile objects (e.g. parked vehicles) and background instances (e.g. trees and buildings), i.e. appearance clusters are labeled as mobile when they have at least 5 percent dynamic objects.
> The instances of these mobile appearance clusters form together the set of (class-agnostic) mobile objects and can be used to train any 3D object detector in an unsupervised manner.
> Compared to existing methods such as OYSTER [1] and LISO [2], we do not rely on computationally expensive self-training to detect static mobile objects because we can discover them entirely unsupervised using their visual appearance.
> This significantly reduces the training time of the detector as only a single training is required instead of multiple training rounds (e.g. sequentially training 5-10 times) while at the same time obtaining a much better detection performance **(see Table 1 in our rebuttal PDF)**.
>
> **Different image encoders**
>
> See global rebuttal text: 'Different image encoders'.
>
> **Ablation for number of pseudo-classes $K_2$ in multi-class detection**
>
> We have added three hyperparameter configurations for the multi-class detection, namely 5, 15, and 20 pseudo-classes **(see Table 2 in the rebuttal PDF)**.
> The table shows that UNION with 5 pseudo-classes achieves the highest average precision (AP) and nuScenes detection score (NDS).
> In general, the multi-class versions of UNION get a slightly lower AP for the vehicle class than 'UNION+Size prior' but a significantly higher AP for the pedestrian class.
> As a result, multi-class UNION gets a relatively high AP for both the vehicle and pedestrian class compared to the baselines using the size prior.
>
> **True positive metrics for nuScenes**
>
> We have added the average translation error (ATE), average scale error (ASE), average orientation error (AOE), and average velocity error (AVE) to **Table 1 in the rebuttal PDF**.
> Note that we set the average attribute error (AAE) equal to 1 by default for the task of class-agnostic object detection because object classes have different attributes in nuScenes so attributes lose their meaning when all mobile classes are combined into a single class.
> Therefore, this error is not shown in the table.
> We use the official nuScenes formula for calculating the nuScenes detection score (NDS).
>
> **Dealing with dynamic objects when aggregating point clouds**
>
> For a frame, we aggregate the past $M$ and future $M$ non-ground point clouds, i.e. $2M+1$ LiDAR scans are aggregated into a single coordinate system, to obtain a denser point cloud.
> Consequently, the spatial clustering may create a single spatial cluster for dynamic objects that are close to each other during this time interval.
> We use $M=7$ for our experiments, resulting in a time interval of 0.70 second, and notice that it rarely happens that different objects are merged into a single spatial cluster.
> A potential solution for this would be to estimate the motion vector for each point of the spatial cluster (scene flow), and determine whether the motion across the entire cluster is consistent.
> If this is not the case, a cluster may be split into multiple spatial sub-clusters.
> However, this heavily relies on an accurate scene flow estimation.
>
> **Cyclist performance for multi-class detection**
>
> From the pseudo-classes of multi-class UNION, there are 1, 1, 3, and 4 pseudo-classes assigned to the cyclist class for 5, 10, 15, and 20 pseudo-classes, respectively.
> So, it is not the case that all pseudo-classes are assigned to either the vehicle or pedestrian class.
> However, the nuScenes evaluation protocol discards the precision-recall curve where the precision or recall is lower than 10 percent.
> As a result, the average precision for the cyclist class for multi-class UNION is equal to 0, while cyclists are detected.
> Therefore, we also evaluated without clipping the precision-recall curve **(see the last column in Table 2 in the rebuttal PDF)**.
> The results show that UNION with 20 pseudo-classes significantly outperforms the baselines with the size prior.
>
> **Potential solutions for improving less frequent objects**
>
> UNION assumes that objects are mobile (i.e. have the potential to move) and that mobile objects with a similar appearance occur relatively often.
> The appearance clusters are made using K-means, meaning that a mobile object type that rarely occurs will most likely be part of an appearance cluster belonging to objects with a dissimilar appearance.
> As a result, the mobile objects may be considered non-mobile.
> Some directions to improve this are (1) exploring other clustering mechanisms than K-means for creating appearance clusters and (2) creating learnable appearance embeddings (self-supervised) that split mobile and non-mobile objects in the appearance feature space better.
>
> [1] Zhang et al. (2023). Towards unsupervised object detection from lidar point clouds. In CVPR.
>
> [2] Baur et al. (2024). LISO: Lidar-only Self-Supervised 3D Object Detection. In ECCV.

---

> > ### Comment · Reviewer_Gw7r · 2024-08-12
> >
> > Thanks for the authors' response, and I have no further concerns.

---

### Official Review · Reviewer_ZwY5 · 2024-07-11

**Soundness:** 2
**Presentation:** 3
**Contribution:** 3
**Rating:** 5
**Confidence:** 4

**Summary:**

The paper introduces UNION, an unsupervised 3D object detection method designed to detect both static and dynamic objects without manual labels. UNION utilizes spatial clustering and self-supervised scene flow to generate object proposals and employs visual appearance encoding to distinguish between static and dynamic objects. This approach enables a single training process, enhancing performance while reducing computational costs. Experiments on the nuScenes dataset show that UNION significantly improves the state-of-the-art performance in unsupervised object discovery, doubling the average precision score to 33.9.

**Strengths:**

1. The approach is quite novel, successfully eliminating the dependence on self-training.
2. The presentation is clear and easy to understand.
3. The paper effectively highlights the similarities and differences with related literature.

**Weaknesses:**

The experiments are relatively limited.
1.  Besides DINOv2, how do other image encoders affect the model's performance?
2. In Table 4, the cyclist AP is 0. Is there a more detailed investigation into the reasons for this? Given that, as a small object, the pedestrian class shows a significant improvement in AP with UNION-10pc compared to UNION+Size prior, why does the cyclist class not exhibit a similar improvement?
3. In Fig. 3, 5% was chosen as the threshold. Why was this value selected? The authors should provide more discussion or experimental support for this choice.

**Questions:**

See weaknesses.

**Limitations:**

The authors acknowledge the limitations of their work and plan to address these in future research.

---

> ### Author Rebuttal · Authors · 2024-08-07
>
> We thank you for your valuable feedback.
>
> **Different image encoders**
>
> See global rebuttal text: 'Different image encoders'.
>
> **Cyclist performance for multi-class detection**
>
> From the pseudo-classes of multi-class UNION, there are 1, 1, 3, and 4 pseudo-classes assigned to the cyclist class for 5, 10, 15, and 20 pseudo-classes, respectively.
> So, it is not the case that all pseudo-classes are assigned to either the vehicle or pedestrian class.
> However, the nuScenes evaluation protocol discards the precision-recall curve where the precision or recall is lower than 10 percent.
> As a result, the average precision for the cyclist class for multi-class UNION is equal to 0, while cyclists are detected.
> Therefore, we also evaluated without clipping the precision-recall curve **(see the last column in Table 2 in the rebuttal PDF)**.
> The results show that UNION with 20 pseudo-classes significantly outperforms the baselines with the size prior.
>
> **Threshold of 5 percent for the fraction of dynamic instances X**
>
> As stated in the global response of the rebuttal, we have further improved the self-supervised scene flow component's recall by also estimating the velocity for spatial clusters with relatively few LiDAR points, and we now use only 20 appearance clusters for filtering the non-mobile instances.
> This improved recall makes the distinction between non-mobile and mobile appearance clusters more obvious **(see Figure 1 in the rebuttal PDF)**.
> In this figure, it can be visually observed that the first 13 clusters (indicated in blue) all have a relatively low dynamic instance fraction compared to the other 7 clusters (indicated in orange), i.e. the fraction of cluster 14 (considered a mobile cluster) is roughly three times as high as the fraction of cluster 13 (considered a non-mobile cluster).
> As a result of this observed difference, we selected 5 percent for splitting the fractions into two groups: (1) clusters with very few dynamic instances (non-mobile clusters) and (2) clusters with many dynamic instances (mobile clusters).
> In addition, we empirically observed that the detection performance is very robust when using the 5 percent threshold while varying, for example, the number of appearance clusters, or when the threshold itself is changed too to, for example, 7.5 percent.
> The intuition is that there should be a clear distinction between the dynamic instance fraction of non-mobile clusters and mobile clusters because non-mobile instances do not have the potential to be dynamic.
> In contrast, mobile instances move very often, as most are part of the traffic.
> There are multiple reasons why the non-mobile clusters do have some dynamic instances, including (1) false positives from the scene flow component, i.e. static objects are estimated to be dynamic (e.g. dynamic tree), and (2) some instances may be part of an appearance cluster while having an appearance dissimilar to the other instances in the appearance cluster because the instance is close to the border of the appearance cluster in feature space.
> Note that using a threshold equal to 0 percent, i.e. just using all object proposals, is equal to the output of the scene flow component and achieves an average precision (AP) of less than half of the performance of UNION (see Table 3 in our submission).
> So the appearance clustering component is essential for achieving high performance.

---

> ### Comment · Reviewer_ZwY5 · 2024-08-13
>
> I have read all the reviews and the responses of the authors. I appreciate the author for providing these experiments. My concern is addressed. So I keep my rating.

---

### Official Review · Reviewer_LFQD · 2024-07-12

**Soundness:** 3
**Presentation:** 2
**Contribution:** 3
**Rating:** 6
**Confidence:** 4

**Summary:**

This paper introduces UNION, a novel method for unsupervised 3D object detection that leverages object appearance-based pseudo-classes. This paper addresses the challenge of training object detection models without manual annotations by using spatial clustering and self-supervised scene flow to generate static and dynamic object proposals from LiDAR data. It then encodes the visual appearances of these proposals to distinguish static objects in the foreground and background. The main contribution is the method design that jointly uses camera, LiDAR, and temporal information to train existing 3D object detectors in an unsupervised manner.

**Strengths:**

1.	The paper is well-structured, with a clear abstract, introduction, methodology, experiments, and conclusion sections that logically flow from one to the next.
2.	The use of figures and diagrams, such as Figure 1, effectively illustrates the process and contributes to the clarity of the UNION method.
3.	The use of pseudo-classes based on object appearance for training classifiers is innovative, offering a new way to tackle multi-class object detection without relying on manual annotations.
4.	The UNION method introduces a new approach to unsupervised learning by combining spatial clustering, self-supervised scene flow, and visual appearance encoding in a synergistic manner. This represents a creative fusion of existing ideas applied to the problem of 3D object detection.

**Weaknesses:**

1.	The claim of being 'the first to do unsupervised multi-class 3D object detection' requires scrutiny. It's noted that other works, such as the one by Wu et al. presented at CVPR 2024, also delve into unsupervised multi-class 3D object detection. If the paper is accepted, it would be prudent to revise this statement to reflect the current state of research accurately and avoid overstating the novelty of the approach.

2.	The decision to solely utilize the nuScenes dataset for experiments raises questions about the breadth of the evaluation. The Waymo dataset, with its denser point clouds, might offer a better testbed for supervised tasks. Furthermore, the official multi-class metric of the Waymo dataset, which assesses the detection of vehicles, pedestrians, and cyclists, provides a more straightforward comparison to fully supervised detectors. It would be beneficial to consider a comparative analysis using the Waymo dataset to strengthen the paper's findings.
Rebuttal may not have time for this work, but can be part of the future.

3.	The observed zero AP for the cyclist class is concerning and warrants a deeper investigation. It is suggested that the discussion should be expanded to address potential solutions to this issue. While the less sample is a contributing factor, it's also essential to explore other possibilities, such as the high similarity in appearance between pedestrians and cyclists, which might lead to misclassification and the pedestrian labels actually being cyclist labels.

[1] Wu, Hai, et al. "Commonsense Prototype for Outdoor Unsupervised 3D Object Detection." Proceedings of the IEEE/CVF Conference on Computer Vision and Pattern Recognition. 2024.

**Questions:**

See weakness.

**Limitations:**

The authors have provided a discussion on the limitations of their work. They acknowledge that the method makes implicit assumptions about the occurrence frequency of objects.

---

> ### Author Rebuttal · Authors · 2024-08-07
>
> We thank you for your valuable feedback.
>
> **First method to do unsupervised multi-class 3D object detection**
>
> Our submission did not compare to [8] as it was not peer-reviewed back then.
> Now that it has been published at CVPR'23, we will add the paper to our related work section and Table 1 (the overview) to accurately reflect the current state of the research field.
> This concurrent work does not use the nuScenes dataset, and their code has not yet been fully released, according to their GitHub page, so we cannot compare with them.
> Wu et al. use multi-class bounding box templates based on Wikipedia to obtain a set of high-quality bounding boxes used for bounding box refinement, i.e. re-size and re-location.
> However, there is a fundamental difference between UNION and their methods of how the pseudo-classes are used to supervise the detector, i.e. class-agnostic versus multi-class detection.
> They do not train object classification as their trained detector only outputs class-agnostic bounding boxes, i.e. they do class-agnostic object detection.
> In contrast, we not only have a class-agnostic object detection experiment (experiment 1) but also have a multi-class experiment as clarified in the global rebuttal text (experiment 2).
> In addition, we do not assume any class-specific geometry prior during training and inference.
> For our multi-class object detection, we create $K_2$ appearance-based pseudo-classes as targets for training object classification and train CenterPoint in a multi-class fashion.
> We can assign the learned pseudo-classes to real classes after training, i.e. during inference, as we match the appearance prototype of each pseudo-class with an example appearance of the real objects that are relevant during inference, such as vehicles, pedestrians, and cyclists.
> Therefore, we are actually the first method to train unsupervised multi-class 3D object detection.
> In our paper, we will explain more extensively how our contribution differs from the existing methods, such as [8].
>
> **Results on Waymo**
>
> We agree that results on the Waymo dataset [9] would improve the breadth of the evaluation.
> However, we do not have enough time during the rebuttal to provide these as Waymo is a very large dataset, and it would take more than one week in total to run the UNION pipeline plus training CenterPoint.
> Therefore, we consider Waymo results as future work.
>
> **Cyclist performance for multi-class detection**
>
> From the pseudo-classes of multi-class UNION, there are 1, 1, 3, and 4 pseudo-classes assigned to the cyclist class for 5, 10, 15, and 20 pseudo-classes, respectively.
> So, it is not the case that all pseudo-classes are assigned to either the vehicle or pedestrian class.
> However, the nuScenes evaluation protocol discards the precision-recall curve where the precision or recall is lower than 10 percent.
> As a result, the average precision for the cyclist class for multi-class UNION is equal to 0, while cyclists are detected.
> Therefore, we also evaluated without clipping the precision-recall curve **(see the last column in Table 2 in the rebuttal PDF)**.
> The results show that UNION with 20 pseudo-classes significantly outperforms the baselines with the size prior.
>
> [8] Wu et al. (2024). Commonsense Prototype for Outdoor Unsupervised 3D Object Detection. In Conference on Computer Vision and Pattern Recognition (CVPR).
>
> [9] Sun et al. (2020). Scalability in perception for autonomous driving: Waymo open dataset. In Conference on Computer Vision and Pattern Recognition (CVPR).

---

> > ### Comment · Reviewer_LFQD · 2024-08-08
> >
> > I read the comments of other reviewers and the author's reply. The author also addressed my concerns. So I keep my previous recommendation.

---

### Author Rebuttal · Authors · 2024-08-06

We thank the reviewers (R1:LFQD, R2:ZwY5, R3:Gw7r, R4:JvDj) for their valuable and detailed feedback.
We appreciate that our method UNION was overall well-received e.g. 'represents a creative fusion of existing ideas applied to the problem of 3D object detection' (R1), 'is quite novel' (R2), and 'has significant improvement on the accuracy compared to existing unsupervised methods' (R4).
In addition, we are pleased that the paper is considered 'well-structured' (R1), and that it 'effectively highlights the similarities and differences with related literature' (R2) and 'studies a significant problem in unsupervised 3D object discovery' (R3).

**Clarification on primary and secondary  contributions (class-agnostics discovery,  multi-class detection)**

OYSTER [1] and LISO [2] only provide class-agnostic pseudo-bounding boxes for training (i.e. no pseudo-class labels).
Our first step is similar in that it provides class-agnostic bounding boxes for mobile objects.
Object proposals are clustered based on appearance embedding into $K_1$ appearance clusters, and mobile objects are obtained by selecting the appearance clusters with at least $X$ percent dynamic instances.
Unlike OYSTER and LISO, our approach does not require computationally intensive self-training iterations.
This is our primary contribution, described in Section 3 (lines 185-252) and evaluated in Section 4.3.

In a second step, however, we extend class-agnostic 3D object discovery to multi-class 3D object detection.
We do so by splitting the obtained set of (class-agnostic) mobile objects into $K_2$ appearance-based pseudo-classes, using the same object appearance embedding technique used in the first step.
During inference, we assign each pseudo-class to a real class using the appearance prototype of the pseudo-class and the appearance of the real classes, requiring a single appearance embedding per real class **(see Figure 1 in rebuttal PDF)**.
Note that this step requires negligible supervision as we do one-shot association during inference.
This is our secondary contribution, described in Section 3 (lines 252-255) and Section 4 (lines 336-345), and evaluated in Section 4.4.
We cannot compare to OYSTER and LISO for this task as they do not have pseudo-classes, i.e. they only have class-agnostic pseudo-bounding boxes, and their source code is not released.
We will adapt the paper text to make the above points more clear.

**Improved results**

We meanwhile improved the results for class-agnostic 3D object detection **(see updated Table 1 in the rebuttal PDF)**.
Previously, we did not use the velocity direction of dynamic objects to correct the orientation of the fitted bounding boxes and did not use the estimated velocity during training (velocity prediction is part of the nuScenes detection score (NDS)).
In addition, we tuned the scene flow estimation, resulting in better performance, i.e. significantly higher recall with slightly lower precision for determining whether an object is dynamic.
We also did a parameter optimization on $K_1$ (grid search) and now use 20 appearance clusters instead of 50.
This all combined increased the average precision (AP) by 4.5 to 38.4, which is more than 3.5 times higher than OYSTER and LISO.
Also, in this rebuttal **(see PDF)**, we present updated results for the multi-class detection, show the bounding box generation pipeline, provide an analysis for the cyclist performance, present results for different image encoders, and provide a runtime analysis for the entire UNION pipeline.

**Different image encoders**

In our submission, all experiments were conducted with a large vision transformer (ViT-L/14) trained using DINOv2 [3] with registers [4].
Table 3 compares UNION's class-agnostic 3D object detection performance for DINOv2 and I-JEPA [5].
DINOv2 uses contrastive learning, which ensures that the representations of different views of the same image are similar while the representations of different images are distinct, while I-JEPA predicts the representations of part of an image from the representations of other parts of the same image.
DINOv2 can process high-resolution images, such as the camera images from nuScenes.
In contrast, I-JEPA can only process square-shaped images of a maximum of 448 by 448 pixels.
As a result, the obtained feature maps from I-JEPA have a lower resolution than the ones from DINOv2.
From Table 3, it can be seen that UNION with DINOv2 outperforms UNION with I-JEPA by 15.6 in AP and 8.4 in NDS.
Please note that our main paper contributions do not depend on specific canonical steps of the UNION pipeline (e.g., image encoding, scene flow estimation). If better approaches become available, UNION can incorporate them.

**Outperforming shelf-supervision**

Recently, a new method named CM3D [6] was released on arXiv; this method has not been peer-reviewed yet.
CM3D is a 'shelf-supervised' 3D object detection method, i.e. off-the-shelve foundation models that were trained with manual labels such as SAM [7] are used, and it achieves an AP and NDS of 27.9 and 25.0, respectively.
In contrast, UNION is *entirely* unsupervised and outperforms CM3D by more than 10 points in AP and more than 5 points in NDS **(see Table 1 in the rebuttal PDF)**, demonstrating the effectiveness of the UNION pipeline, while not relying on manually labeled data sources.

[1] Zhang et al. (2023). Towards unsupervised object detection from lidar point clouds. In CVPR.

[2] Baur et al. (2024). LISO: Lidar-only Self-Supervised 3D Object Detection. In ECCV.

[3] Oquab et al. (2024). DINOv2: Learning robust visual features without supervision. In TMLR.

[4] Darcet et al. (2024). Vision Transformers Need Registers. In ICLR.

[5] Assran et al. (2023). Self-supervised learning from images with a joint-embedding predictive architecture. In CVPR.

[6] Khurana, et al. (2024). Shelf-Supervised Multi-Modal Pre-Training for 3D Object Detection. arXiv.

[7] Kirillov et al. (2023). Segment anything. In ICCV.

---

### Decision · Program_Chairs · 2024-09-25

**Decision:**

Accept (poster)

**Comment:**

This paper proposes an unsupervised object detection (or object discovery) method using 3D data. The input consists of multimodal data such as images and LiDAR point clouds, while the output is a set of pseudo-bounding boxes and pseudo-class labels of objects. One of the significant contributions of the proposed method is the elimination of time-consuming iterative self-training, which is commonly used in previous methods. The method effectively utilizes multimodal data and temporal information to distinguish between static and dynamic objects. The paper's contributions and comparisons with related work are very clearly stated and are presented in an easily understandable manner using figures and tables. The clarity of the paper, the novelty of the method, and its high performance were recognized by most reviewers, all of whom gave ratings leaning towards acceptance. Although there were some questions regarding missing experimental results, these were addressed well with additional experimental results in the authors' rebuttal, satisfying all reviewers and leading them to maintain their initial evaluations. Based on these points, it can be concluded that this paper is suitable for acceptance at this conference.